# Elucidation of Volatiles, Anthocyanins, Antioxidant and Sensory Properties of *cv*. Caner Pomegranate (*Punica granatum* L.) Juices Produced from Three Juice Extraction Methods

**DOI:** 10.3390/foods10071497

**Published:** 2021-06-28

**Authors:** Jurga Budiene, Gamze Guclu, Kouame Fulbert Oussou, Hasim Kelebek, Serkan Selli

**Affiliations:** 1Department of Organic Chemistry, State Research Institute Center for Physical Sciences and Technology, Sauletekio Ave. 3, LT-10222 Vilnius, Lithuania; j.budiene@gmail.com; 2Department of Food Engineering, Faculty of Agriculture, Cukurova University, Adana 01130, Turkey; gguclu@cu.edu.tr (G.G.); oussoufulbert@gmail.com (K.F.O.); 3Department of Food Engineering, Faculty of Engineering, Adana Alparslan Turkes Science and Technology University, Adana 01250, Turkey; hkelebek@atu.edu.tr; 4Department of Nutrition and Dietetics, Faculty of Health Sciences, Cukurova University, Adana 01250, Turkey

**Keywords:** pomegranate, *cv*. Caner, juice extraction methods, aroma compounds, anthocyanins, antioxidants

## Abstract

This study deals with the characterization of the phytochemical profiles and antioxidant activities of cv. Caner pomegranate (*Punica granatum*) juices obtained from three different juice extraction methods including halved pomegranate (HPJ), arils (AJ), and macerated arils (MAJ) extraction for the first time. It was found that the type of the juice extraction process had substantial effects on the volatiles, anthocyanin compositions, and antioxidant activities of the samples. Results showed that the AJ sample (593 mg L^−1^) had more anthocyanin compounds followed by HPJ (555 mg L^−1^) and MAJ (408 mg L^−1^) samples. GC-MS analysis revealed a total of 34 volatile compounds. The highest number of volatiles was found in the MAJ sample (1872 µg L^−1^); thus, the aril maceration process played an important role in increasing the volatiles as compared to the HPJ (751.8 µg L^−1^) and AJ (710.7 µg L^−1^) samples. Sensory analysis showed that the HPJ sample was the most preferred and its general impression was higher as compared to the AJ and MAJ samples. The findings of this study elucidated that the juice extraction technique had a significant influence on the phytochemical profiles, sensory quality, and antioxidant activity of pomegranate juices.

## 1. Introduction

Pomegranate fruit (*Punica granatum* L.) belonging to the Punicaceae family is primarily considered to be a crucial source of bioactive compounds that are claimed to possess health beneficial properties. Thus, recently, there has been a massive increment in the popularity of pomegranate consumption [1]. This fruit, originating from the Middle East, has been a widely known fruit since ancient times. The total global production of pomegranate is reported to be over 3 million tons, and Turkey is an important producer with a total annual production of 581.189 tons [2]. The interest in pomegranate fruits is growing year by year not only because it is pleasant to eat, but also because it is a fruit with a good source of minerals, acids, sugars, vitamins, polysaccharides, and phenolics such as anthocyanins and phenolic acids [3]. Anthocyanins are well known to function as natural antioxidants and play a role in the protection against oxidative stress, reduction of risks of chronic diseases, as well as prevention of their progression [4,5]. Recently, numerous studies demonstrated that there has been a relation between the intake of fruits and vegetables comprising natural antioxidants and the inhibitions of many diseases including cancers [6,7]. Pomegranate fruits are generally consumed fresh, but lately, there is a substantial demand in industry to obtain pomegranate juice, jams, jelly, vinegar, and wine [8].

The overall quality of pomegranate cultivars depends on its taste components, aroma profiles, and color properties. Among these, aroma is a vital quality criterion for foods affecting the consumer’s acceptance and preference to a higher extent. These compounds, also known as volatile organic compounds, can be chemically classified as aldehydes, alcohols, acids, ketones, esters, lactones, and terpenes. These constituents are also present in very low concentrations in food samples, and due to their different molecular characteristics, every single aroma compound has different contributions to the final aroma of a food sample [9]. Even if exhaustive works are present in the literature concerning the health effects of pomegranates and their juices, only a small portion of them focus on volatile compounds. In one of the most comprehensive papers, it is reported that a mixture of volatiles responsible for the green, fruity, floral, and earthy notes composes the pomegranate juice aroma [10]. In addition, Beaulieu et al. [11] investigated the quality properties and aroma components in sweet, sweet–sour, and sour pomegranate cultivars from around the world that were grown in a collection of California-grown pomegranates from the National Clonal Germplasm Repository. They found that aldehyde and terpene compounds characterize cultivar differences and 3-hexenol and 1-hexanol were the dominant compounds.

Another important group of compounds in pomegranates are phenolics. They have great importance because of their role for quality parameters such as color, taste, and their favorable impacts on health [1]. Pomegranates are very rich in terms of anthocyanins, ellagic acid, phytoestrogenic flavonoids, and tannins, which have the ability to act as antioxidant properties [12,13]. Among these, anthocyanins are the key pigments in pomegranate cultivars found in various parts of the pomegranate trees, leaves, flowers, peels, and fruits. In the extant literature, six anthocyanin molecules were determined in different pomegranate fruits, including mono- and di-glucosides of cyanidin, delphinidin, and pelargonidin [14]. In addition to their contribution to sensory profiles, they are also important for health with their antitumor, antioxidant, antimicrobial, and anti-inflammatory effects. As pomegranates and their products are rich in phenolics, specifically anthocyanins, they display these health beneficial properties, and [14,15]. Kostka et al. [15] extracted and separated pomegranate polyphenolics into an anthocyanin and copigment fraction with the aid of the adsorptive membrane technique. They elucidated that the total phenolics and free radical scavenging activity were considerably higher in the XAD-7 extract compared to the juice and that anthocyanins and copigments act together in decreasing oxidative stress.

Pomegranate fruit, being rich in these mentioned health-beneficial and bioactive compounds, requires handling with care. The processes like juice extraction to be applied on this fruit may result a loss in desired compounds, so the conditions of the procedures should be decided carefully to get the highest benefit from the compounds. There has been no comprehensive work on the color quality, volatiles, anthocyanins, sugar, and organic acid constituents of *cv*. Caner pomegranate juice produced from different juice extraction methods in the extant literature. So, this work was established to assess the influence of three juice extraction methods (halved fruit, arils, and macerated arils extraction) on the amounts and types of the volatiles, anthocyanins, antioxidant properties, sugars, and organic acids of *cv*. Caner pomegranate. Besides, in the work that was conducted in this comprehensive manner for the first time, a sensory study was also utilized for investigating the influence of three different juice extraction techniques.

## 2. Materials and Methods

### 2.1. Pomegranate Juice Samples

Pomegranate fruits of *cv*. Caner were harvested from the Cukurova University’s experimental orchards located in Adana province of Turkey at the full ripeness stage. Ten kilograms of pomegranates were used for each batch. Fruits were washed with tap water, dried with towels, and divided into two parts. Three different juice extraction methods were employed to obtain the juice samples. Pomegranates from the first batch were cut into two pieces and the juice was obtained by a hydraulic stainless-steel bladder press (Speidel, Bayrakli-Izmir, Turkey) at 2 atm pressure and this sample was called as halved pomegranate juice (coded by “HPJ”). The second sample (coded as “AJ”) was extracted from the arils by pressing at 2 atm pressure using a hydraulic press without crushing the seeds. For the last sample, arils of the fruit were separated from the skin (membrane) manually then they were crushed without damage to their seeds and kept at 4 °C for 3 h for maceration (coded “MAJ”). After maceration, the juice was extracted with the same procedure used for the AJ sample. Juices were frozen and stored at −20 °C until analyses.

### 2.2. General Chemical Analysis

The pH and total soluble solids were determined immediately after obtaining the juice samples with the use of a pH meter (Orion 3 STAR pH Benchtop Meter, Thermo Scientific, Waltham, MA, USA) and a refractometer (Carl Zeiss, Jena, Germany) (expressed as °Brix), respectively. The titratable acidity of the juice samples was determined by titration with NaOH (0,1 N) [16]. The color of juice was quantified using a colorimeter (ColorQuest XE, HunterLab, Reston, VA, USA) and expressed as L* (lightness), a* (redness), and b* (yellowness). The average values of the triplicate measurements were reported.

### 2.3. Analysis of Sugars and Organic Acids

Analysis of sugars and organic acids were carried out according to the method of Lee and Coates [17] with slight modifications. Pomegranate juices were centrifuged (6000× *g*, 15 min), filtered through a 0.45 µm filter, and diluted 1:1 with ultra-pure water. The obtained extract was directly injected to the HPLC system equipped with LC-20AD, SPD-20A UV and RID 10 A detectors (Shimadzu, Kyoto, Japan). HRC NH2 column (Biorad, 150 × 4.6 mm, 5 m) was used in both analyses. The mobile phase consisted of 5 mM H_2_SO_4_ solution with a flow rate of 0.5 mL min^−1^. Glucose and fructose standards (Sigma-Aldrich, St. Louis, MO, USA) were used to obtain a calibration curve for the calculation of sugar concentrations. Organic acid concentrations were quantified in same manner with citric, malic, and ascorbic acid standards (Sigma-Aldrich, St. Louis, MO, USA). The limit of detection (LOD) and limit of quantification (LOQ) for the analysis were calculated at a signal-to-noise ratio (S/N) of about 3 and 10, respectively.

### 2.4. Analysis of Individual Anthocyanins

An Agilent 1100 HPLC system (Agilent Technologies, Palo Alto, CA, USA) operated by ChemStation software was used in the analysis. All juices were centrifuged (6000× *g*, 15 min) and filtered through 0.45 µm filter (Millipore) before injection. The analysis was performed on a Beckman Ultrasphere ODS (Roissy CDG, France; 4.6 mm × 250 mm) column. The mobile phase consisted of two solvents: Solvent A; water/formic acid (95:5; *v/v*) and Solvent B; acetonitrile/solvent A (60:40; *v/v*). Anthocyanins were separated in reference to the method reported in Kelebek and Selli [18]. The compounds were identified using the retention times as spectra were matched to authentic standards. The quantities of different anthocyanins were assessed from the peak areas and calculated as equivalents of representative standard compounds in calibration curves as follows: At 520 nm (anthocyanins), cyanidin 3-glucoside and cyanidin 3-rutinoside, respectively. The results were expressed as mg L^−1^. The limit of detection (LOD) and limit of quantification (LOQ) were calculated at a signal-to-noise ratio (S/N) of about 3 and 10, respectively.

### 2.5. Antioxidant Capacity Analyses

The ABTS (2, 2′-azinobis-(3-ethylbenzothiazoline-6-sulfonic acid) and DPPH (1, 1-diphenyl-2-picrylhydrazyl) assays were used in the determination of the antioxidant capacities of the pomegranate juice samples. In regard the antioxidants, the assays were performed in reference to the method reported in Kelebek et al. [19]. The absorbance of the solution was determined by a UV-VIS Spectrophotometer (UV-1601, Shimadzu, Kyoto, Japan). The absorbance values of the ABTS and DPPH solutions were recorded at 734 and 517 nm, respectively.

### 2.6. Analysis of the Volatile Composition

#### 2.6.1. Extraction of Volatile Compounds

Volatiles of the pomegranate juice samples obtained from three different juice extraction methods (AJ, HPJ, and MAJ) were extracted by the liquid–liquid extraction technique. The procedure was applied with slight modification according to Selli and Kelebek [20]. Dichloromethane was preferred as the solvent in the extraction of the aroma compounds. Briefly, 100 mL centrifuged sample (6000× *g* for 15 min at 4 °C), 40 mL dichloromethane, and 5 µL of 4-nonanol (43.3 mgL^−1^) as an internal standard was stirred at 4 °C for 30 min under nitrogen gas. After stirring, the mixture was centrifuged again (9000× *g*, 15 min at 4 °C) and the organic phase was dehydrated by using anhydrous sodium sulfate. The extract was condensed to 5 mL in a concentrator (Kuderna Danish, Supelco, St. Quentin, France) at 40 °C and thereafter, to 0.2 mL by using a flow of nitrogen [20]. After this stage, each extract was placed in a glass vial of 2 mL with a Teflon-lined cap. Samples were extracted in triplicate.

#### 2.6.2. GC-FID and GC-MS Analysis of Volatile Compounds

The system of gas chromatography (GC) comprised an Agilent 6890 chromatograph interfaced to an FID (flame ionization detector) and an MSD (mass selective detector) (Agilent 5973, Wilmington, DE, USA). To separate the volatiles of the samples, a DB-Wax column was utilized (0.5 µm thickness × 0.25 mm i.d. × 30 m length; J&W Scientific, Folsom, CA, USA). The analysis was applied according to the method used in Keser et al. [21] with a slight modification. Volatiles of the samples were studied based on the retention index and mass spectra on the DB-Wax column using a commercial database of spectra (Wiley 6, NBS 75 k). The volatiles were then quantified utilizing the internal standard with 4-nonanol at 43.3 μg L^−1^. Response factors were computed based on the intensity ratio of each volatile to 4 nonanol and ratios of peak areas were corrected by using each constituent’s response factor [21,22]. Subsequently, the means and standard deviations were computed for the GC analyses in triplicate. Retention index data for the volatiles were subtracted by utilizing the n-alkane series (C8–C32).

### 2.7. Sensory Analysis of Pomegranate Juice Samples

Ten trained panelists (five women and five men aged between 25–47) from Food Engineering Department of Cukurova University (Adana, Turkey) evaluated pomegranate juice samples in terms of overall appearance including color, pulpiness, sourness, sweetness, astringency, fruity, and floral notes. Samples were prepared at room temperature approximately 30 min before the analysis. The juices were served in disposable, odorless plastic cups after shaking. The order of samples was randomized, and the panelists were asked to evaluate the samples on a 10 cm evaluation scale [13].

### 2.8. Statistical Data Analysis

The data were studied by using analysis of variance (ANOVA) in SPSS (v.24.0, SPSS Inc., Chicago, IL, USA). In addition, principal component analysis (PCA) was employed in XLSTAT software (trial version of 2020, Addinsoft, New York, NY, USA).

## 3. Results and Discussion

### 3.1. Physicochemical Properties of Pomegranate Juices

Fruit size, skin, and aril colour are known to directly affect the customer acceptability and these properties depend on variety, climatic, and agricultural conditions. *cv*. Caner pomegranates used in the study were of medium size compared to the samples reported on Tunisian pomegranate cultures [23]. Fruit skin and arils were observed to have an intensive pink-red colour.

The chemical composition of the samples is displayed in Table 1. It was detected that juice extraction methods did not significantly change pH values. Similar results were reported by other researchers [24,25]. The other characteristics responsible for the determination of juice quality are total soluble solids (°Brix) and titratable acidity. The total soluble solid values were between 14 and 16 °Brix in pomegranate samples. The differences in the total soluble solid values may be related to the tannins formed by the damage of rind cells during the juice extraction process in the HPJ sample. The existence of tannins is reported as an important problem in juices extracted from whole fruits. Accordingly, a bitter taste can develop, and this must be removed by industrial processing in order to meet the consumer demand [26]. The AJ and HPJ samples were revealed to have close values of titratable acidity (TA) while juice obtained from the MAJ had a lower TA value. Vazquez-Araujo et al. [27] reported that no significant differences were found in pH, TA, and TSS values between Wonderful pomegranate juice samples from arils with albedo homogenate and from arils only.

Significant differences were observed in the HPJ, AJ, and MAJ samples in terms of all L*, a*, b* colour parameters (*p* < 0.05). Colour is an important food quality parameter, and it influences the consumer’s preference and market value of the final product. As displayed in Table 1, the AJ (L*: 15.7) was found to be darker than HPJ (L*: 22.0) and MAJ (L*: 23.9) samples. The colour of pomegranate juices is known to be affected by the treatments applied in processing. The use of juices with low values of L* parameter (darker red–purple) is recommended in the literature [28,29]. In comparison to the reported L*, a*, b* values in the literature, the colour of the AJ juice sample from the *cv.* Caner pomegranate was found to be almost twice as dark as the juices of nine Spanish pomegranate varieties [30]. In terms of redness (a*) and yellowness (b*), *cv.* Caner pomegranate juices, independently from the extraction methods, were observed to have higher values than both Spanish and Tunisian cultivars (values are in the range of 3.0–29.7 for a* and −1.7–23.7 for b*) [23,30].

### 3.2. Sugars and Organic Acids of the Pomegranate Juice Samples

Sugars were detected as glucose and fructose in the samples (Table 2). Sugar contents varied in the range of 51.0–56.0 g L^−1^ for glucose and 61.2–69.5 g L^−1^ for fructose depending on the juice extraction methods. These findings are in agreement with the sugar contents determined in pomegranate juices in other previous studies also reporting the glucose and fructose as the principal component of the juice sugar contents. Individual sugar contents reported in the literature were 57–65 g L^−1^ for glucose and 60–71 g L^−1^ for fructose in 40 different Spanish cultivars [31] and 58–76 g L^−1^ for glucose and 58–71 g L^−1^ for fructose in the arils from six pomegranate varieties in Turkey [24].

In the present study, citric, malic, and ascorbic acids were found as organic acids in the juice samples (Table 2). Among these, ascorbic acid was present in minor amounts in all samples. These data showed a similar correlation with the previous studies [24,31]. In the present study, the juice sample obtained after maceration (MAJ) had lower amounts of sugars, but the organic acid contents were similar in all three juice samples. When the three samples were compared, it was found that malic and ascorbic acid levels were not significantly changed by the juice extraction methods. The highest amount of citric acid was found in the HPJ sample (14.1 g L^−1^). This high amount of citric acid is thought to have passed from the peel into the juice since pomegranate peel is known to contain a high amount of citric acid [32].

### 3.3. Anthocyanin Compositions of Pomegranate Juice Samples

Anthocyanins are crucial quality compounds as they are responsible for the attractive red colour in pomegranate juices [33]. In *cv.* Caner pomegranate juice samples analyzed in the present study, the total anthocyanin concentration was determined as 408, 555, and 593 mgL^−1^ in the MAJ, HPJ, and AJ samples, respectively (Table 3). Thus, the juice extraction process method had a significant influence on the amount of anthocyanin compounds of the juice samples. HPLC chromatogram of these compounds obtained at 520 nm was displayed in Figure 1.

In the present study, a total of six anthocyanins, typical for pomegranates, were detected including cyanidin-3-glucoside (Cya3), cyanidin-3,5-diglucoside (Cya3,5), delphinidin-3-glucoside (Dp3), delphinidin-3,5-diglucoside (Dp3,5), pelargonidin-3-glucoside (Pg3), and pelargonidin-3,5-diglucoside (Pg3,5) in all juice samples. Cya3,5 was determined to be the dominant anthocyanin in the studied samples followed by Dp3,5 (Table 3). This compound was reported as the main anthocyanin in Georgian pomegranate juices obtained from cultivars of Rose, Afganski, Nikitski ranni, and Fleshman [25]. In addition, Kelebek and Canbas [34] stated that *cv.* Hicaz pomegranate juices from Turkey are also rich in cyanidin-3,5-diglucoside. In the current study, among the anthocyanins, Pg3 had the lowest amount in *cv.* Caner juice samples. Fawole and Opara [35] reported that Cya3,5 was the major anthocyanin followed by Dp3,5 in *cv.* Bhagwa juices similar to the results found in the present study. They also stated that the amounts of Cya3,5 and Dp3,5 increased with ripening stage. With regard to the effects of juice extraction methods on the anthocyanin contents, maceration of arils resulted in a reduction in their total amounts. This decrement may originate from the effects of light exposure, oxygen, storage time, and temperature [36]. Similarly, Galego et al. [37] reported that anthocyanins are highly sensitive pigments and significant changes in their structure are detected even if these compounds are stored at −20 °C.

### 3.4. Antioxidant Capacity of the Juice Samples

Two methods (ABTS^+^ and DPPH^−^) were employed to assess the antioxidant capacities of the pomegranate juices in the present study. These two assays are basically radical scavenging methods based on the decolorization of free radicals [38]. The selection of these two assays was based on the chemical structure of antioxidant substances and the interaction of them with oxidants. ABTS^+^ can be applied to both hydrophilic and lipophilic antioxidants while DPPH^−^ is applicable to hydrophobic systems as it can be dissolved in organic media [38]. Antioxidant activity values were detected in the range of 6.1–11.3 mmol Trolox L^−1^ for ABTS^+^ and 5.8–14.7 mmol Trolox L^−1^ for DPPH^−^ analyses. These results are consistent with previous studies that reported high antioxidant power in pomegranates of different varieties [12,23,39]. In addition, the AJ sample was found to exhibit higher antioxidant activities than aril juices of *cv.* Mollar de Elche pomegranate of Spain [13]. In the current study, juice extraction methods were determined to have an influence on the valuable benefit of antioxidant activity as both assays revealed for all three pomegranate juice samples. The HPJ sample had higher antioxidant activities than the AJ and MAJ samples (Table 3). Similarly, Mphahlele et al. [40] reported that *cv.* Wonderful pomegranate juices obtained from halved fruits showed higher antioxidant activity (1337 µM Trolox mL^−1^) in comparison with the juices from arils and arils plus seeds. With respect to this, the research conducted by Orak et al. [41], which examined different parts of the pomegranate, revealed that the pomegranate peel extract had the highest DPPH scavenging activity compared to the pomegranate juice or seeds. Thus, it can be deduced that the HPJ sample with higher antioxidant activity is related to the phenolics like hydrolysable tannins, punicalagins, anthocyanins, and phenolic acids found in non-edible parts of pomegranates (peel, carpellary membranes, etc.). Additionally, with regards to maceration, it is observed that the process had an at least two-fold diminishing effect on the antioxidant activity of pomegranate juice. Such decrease in the antioxidant activity of the MAJ sample may be due to the degradation of the anthocyanins, which is more probable during maceration.

### 3.5. Volatile Composition of the Pomegranate Juice Samples

Pomegranate juices have been reported to have low quantities of volatiles leading to a less intense aroma [30]. To investigate the effects of different juice extraction methods on the aromatic profile of *cv.* Caner pomegranate juice samples, liquid–liquid extraction was performed to obtain volatile compounds from three juice samples in the present study. A total of 34 volatile compounds including mainly alcohols, esters, and terpenes were identified and quantified as in Table 4. The total aroma composition of the samples was determined in the range of 710.7 and 1872.0 µgL^−1^ with the MAJ having the highest volatile concentration. These results infer that juice extraction processes had a substantial effect on both concentration and composition of the volatile compounds.

Alcohols were the dominant chemical group both quantitatively and qualitatively in the AJ and MAJ samples while ketones were detected in higher amounts in HPJ sample. 1-Hexanol was predominant in aril and macerated juices with amounts ranging 185–508 µgL^−1^ whereas 2,6-di(t-butyl)-4-hydroxy-4-methyl-2,5-cyclohexadien-1-one had the highest concentration with a value of 405 µgL^−1^ in the MAJ sample. 1-Hexanol is a six-carbon aliphatic alcohol generally found in different parts of pomegranates and their products [35,42,43,44]. C6 alcohols are generally produced by the lipoxygenase-hydroperoxide lyase metabolic pathways. This compound was reported to contribute to green, grass, and fruity attributes in pomegranate juices [43,45]. The amount of 1-hexanol was found to be higher in the AJ (185 µgL^−1^) than the HPJ (53.5 µgL^−1^) juice sample while the MAJ had the highest value (508 µgL^−1^). Similarly, Mphahlele et al. [40] determined that juices obtained from arils plus seeds had higher 1-hexanol concentration than juices obtained from whole or halved fruits. In another study, maceration of juice with *Arbutus unedo* L. distillate into pomegranate liquors caused increases in the amount of 1-hexanol level from 1.14 to 5.58 mg 100 mL^−1^, which was originally absent in the distillate. These literature data are in agreement with our results in the way that the juice exposed to maceration exhibited the highest concentration of this compound. Another important six-carbon alcohol was determined to be (*Z*)-3-hexen-1-ol in all three juice samples in the present study. This compound was also found in relatively higher amounts in other pomegranate juices providing a strong fresh green odor due to its higher odor activity value [46]. This compound was also determined by Vazquez-Araujo et al. [47] in fresh and commercial pomegranate juices from the USA.

Ketones were the second crucial group of volatiles in all samples in the current study. Among them, 2,6-di(t-butyl)-4-hydroxy-4-methyl-2,5-cyclohexadien-1-one was detected for the first time in pomegranate juice in this current study as volatile compound to the best knowledge of the authors. It was also reported to exist in fermented soy sauce and white strawberry (*Fragaria chiloensis*) and characterized by caramel odor [48,49]. On the other hand, 2-Nonanone was detected only in the HPJ sample. This was also reported to be an odor active compound responsible for cheesy, fruity, floral, and green attributes in pomegranate samples [43,46]. In addition, this compound was detected in *cv*. Wonderful pomegranate juices obtained with the peels [43].

Other important aromatic compounds in the juice samples were found to be terpenes and esters (Table 4). Terpenes are secondary metabolites synthesized from isoprene units. Three terpenes were detected including linalool, *α*-terpineol, and *p*-cymen-8-ol while *α*-terpineol was common in all three samples. This compound was also found in Chinese, American, South African, and other Turkish pomegranate juices displaying floral and fruity attributes [29,43,46,50]. Besides *α*-terpineol, linalool and *p*-cymen-8-ol were also reported to be detected in minor levels in different *cvs*. pomegranate juices [45,46,47,48,49,50,51,52]. With regard to esters, in the present study, (*Z*)-3-hexenyl acetate and isopulegol acetate were detected in minor amounts. In previous studies, (*Z*)-3-hexenyl acetate was reported to be formed as a result of the conversion of its corresponding six-carbon alcohols [53]. This compound was also previously reported in different Spanish pomegranate juices and attributed to the green, sweet, and banana odor notes [28,52].

Butyrolactone was the only lactone compound found in all juice samples in the present study. This lactone is responsible for sweet caramel odor and was also reported to be found in *cv.* Wonderful pomegranates [43]. In the current study, *β*-Ionone is a norisoproneoid typical to berries and was found only in trace levels in aril and halved pomegranate juice samples. This compound was also reported to be found in some Spanish pomegranate juices by Vazquez-Araujo et al. [47].

### 3.6. Sensory Analysis

Nine sensory attributes were utilized to describe the flavor of pomegranate juice samples: Pulpy, aroma, color, sourness, sweetness, astringency, fruity, floral notes, and general impression. Mean scores for each descriptor are presented in the spider web diagram in Figure 2. Pulpy was an attribute present in all samples ranging from 3.8 and 5.6. This finding is similar to the data presented by Vazquez-Araujo et al. [47] for pomegranate juice macerated with albedo homogenate. The sweet, sour, floral, and astringency attributes of the samples were found to be very similar and no statistically significant difference was found. HPJ (halved pomegranate juice) samples were characterized by more intense fruity and floral odors. Floral note values were found to be the lowest among all sensory attributes (3.0–3.4) similar to the data reported by Vazquez-Araujo et al. [47]. The largest statistically significant differences (*p* < 0.05) were for aroma, color, and general impression attributes in the present study. The HPJ sample was less pulpy, but it had high scores in all other attributes as compared to the other two samples, and it was also determined to be the most preferred sample by panelists with the highest general impression score.

### 3.7. Principal Component Analysis (PCA) of the Juice Samples

Principal component analyses were also performed on the physicochemical and volatile composition of the juice samples to obtain a better observation of the variances between juice samples obtained from three different juice extraction methods (Figure 3). PCA biplots are displayed for the physicochemical and volatile compositions in Figure 3a,b, respectively. Two PCA components were sufficient to explain 100% of the total variability for both the physicochemical data (F1 = 76.0% and F2 = 24.0%) and for the volatile compositions (F1 = 54.4% and F2 = 45.6%). In these plots, Figure 3a investigates the differences in juices in terms of physicochemical properties while the latter exhibits the dispersion of volatiles between samples. In both figures, it can be seen that the AJ, HPJ, and MAJ juice samples were distinguished efficiently. The HPJ sample was characterized by total sugar, DPPH, ABTS, TSS, total organic acid, and anthocyanins like Dp3 and Cya3. While aril juice (AJ) is seen to be generally categorized by anthocyanins such as Cya3,5, Pg3 and Pg3,5; the MAJ sample was located closely to pH values. With regard to volatile composition, juice samples were discriminated successfully as seen in Figure 3b. MAJ and HPJ samples were located on the positive F1 axis while AJ sample was placed in the negative side. The MAJ sample was categorized by volatile compounds like linalool, (*E*)*-* and (*Z*)-3-hexen-1-ol; 2-nonanone and some phenols were located close to the MAJ and compounds like nonanal and methylacetoin were found to be related with the AJ sample in the biplot.

## 4. Conclusions

This study was the first detailed approach to elucidate volatiles, anthocyanins, sugars, organic acids, and antioxidant properties of juices of Caner cultivar pomegranate fruits obtained from three different juice extraction methods (halved pomegranate juice, HPJ, aril juice, AJ, and macerated aril juices, MAJ). The experimental results clearly showed that these compounds in all juice samples were significantly altered by the applied juice extraction method.

The total quantity of the aroma compounds was detected to be increased as a result of the macerating process (MAJ sample). 1-Hexanol and (*Z*)-3-hexen-1-ol were the main volatiles in all juice samples. When the three juice samples were compared in terms of volatiles, it was found that MAJ was quite rich in terms of these compounds compared to the HPJ and AJ samples. Among the volatile compounds, 2,6-di(t-butyl)-4-hydroxy-4-methyl-2,5-cyclohexadien-1-one was determined for the first time in this study as volatile compounds in pomegranate juices to the best knowledge of the authors. The main anthocyanins were found as cyanidin-3,5-diglucoside followed by delphinidin-3,5-diglucoside in all three juice samples. In contrast to the volatiles, it was found that all of the anthocyanin components of the MAJ sample were significantly decreased during maceration due to the degradation of these compounds. Sensory analysis revealed that the general impression of the HPJ sample was higher as compared to the AJ and MAJ samples.

## Figures and Tables

**Figure 1 foods-10-01497-f001:**
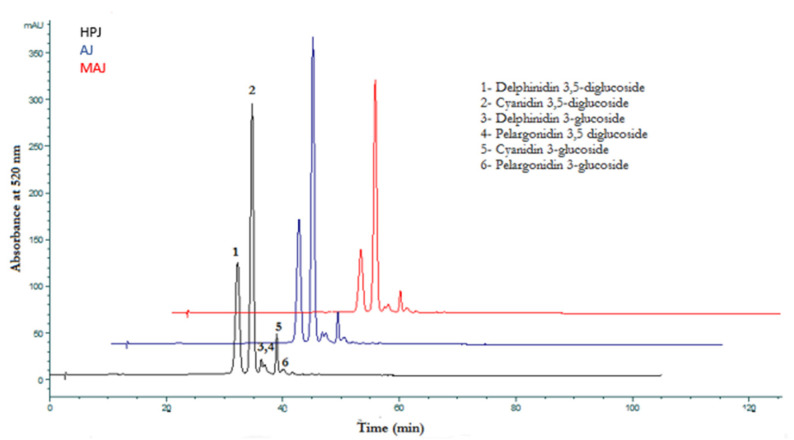
HPLC chromatograms of anthocyanins identified in pomegranate juice samples. (AJ—Arils juice, HPJ—halved pomegranate juice, MAJ—macerated arils juice).

**Figure 2 foods-10-01497-f002:**
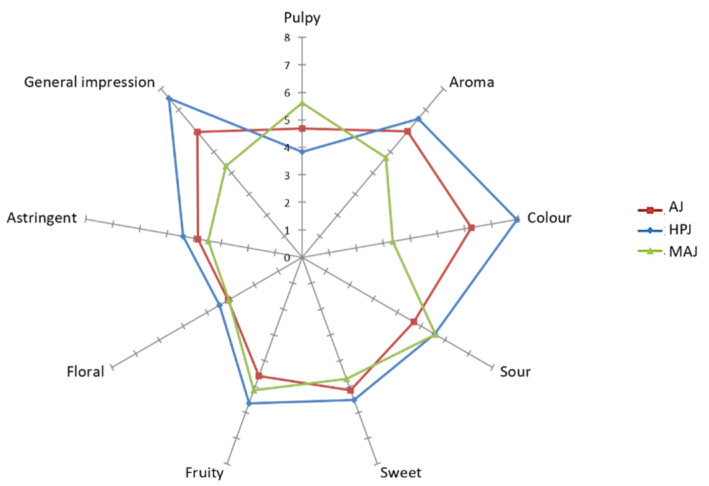
Sensory profiles of pomegranate juice samples. (AJ—Arils juice, HPJ—halved pomegranate juice, MAJ—macerated arils juice).

**Figure 3 foods-10-01497-f003:**
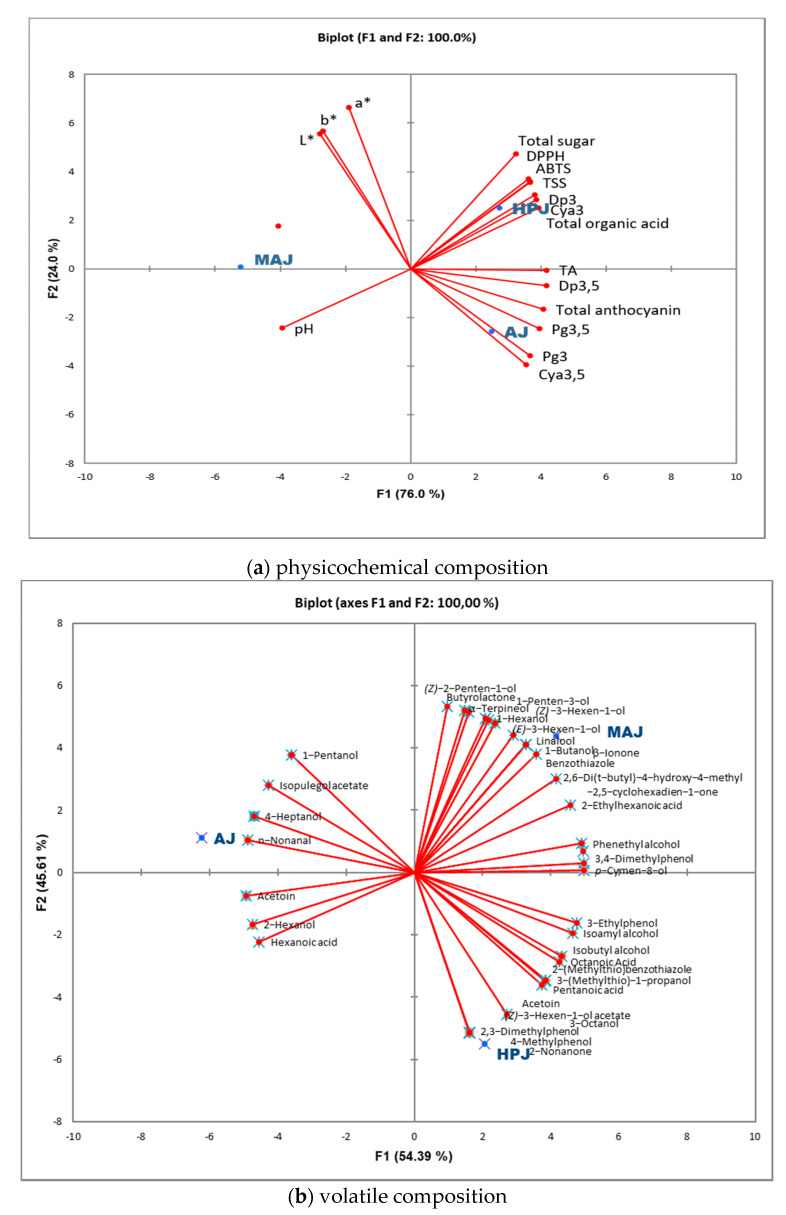
PCA biplots of (**a**) physicochemical (first plot) and (**b**) volatile composition (second plot) of pomegranate juice samples (AJ—arils juice, HPJ—halved pomegranate juice, MAJ—macerated arils juice).

**Table 1 foods-10-01497-t001:** Standard chemical analysis of the pomegranate juice samples.

Sample Name *	AJ	HPJ	MAJ
pH	3.1 ± 0 ^a^	3.0 ± 0.1 ^a^	3.1 ± 0.1 ^a^
TA (g citric acid L^−1^)	16.7 ± 0.2 ^a^	16.9 ± 0.1 ^a^	13.7 ± 0.1 ^b^
TSS (°Brix)	15.0 ± 0 ^b^	16.0 ± 0 ^a^	14.0 ± 0 ^c^
Colour			
L*	15.7± 0.1 ^c^	22.0± 0.1 ^b^	23.9± 0 ^a^
a*	45.0± 0.1 ^b^	52.9± 0.1 ^a^	52.7± 0.1 ^a^
b*	26.9± 0. 1 ^c^	37.6± 0.1 ^b^	40.6± 0.1 ^a^

* AJ—arils juice, HPJ—halved pomegranate juice, MAJ—macerated arils juice. Different letters (a, b, c) on the numbers in same row indicate significant differences (*p* < 0.05) between the juice samples.

**Table 2 foods-10-01497-t002:** Sugars and organic acids (g L^−1^) found in the pomegranate juice samples.

Samples	Glucose	Fructose	Citric Acid	Malic Acid	Ascorbic ACID
AJ *	52.2 ± 0.4 ^b^	64.4 ± 2.3 ^b^	12.3 ± 0.3 ^b^	12.1 ± 0.1 ^a^	1.3 ± 0.1 ^a^
HPJ	56.0 ± 1.1 ^a^	69.5 ± 3.3 ^a^	14.1 ± 0.3 ^a^	12.2 ± 0.4 ^a^	1.6 ± 0.2 ^a^
MAJ	51.0 ± 1.4 ^b^	61.2 ± 2.7 ^c^	10.2 ± 0.1 ^c^	10.5 ± 0.4 ^a^	1.3 ± 0.4 ^a^

* AJ—arils juice, HPJ—halved pomegranate juice, MAJ—macerated arils juice. Different letters (a, b, c) on the numbers in same column indicate significant differences (*p* < 0.05) between the juice samples.

**Table 3 foods-10-01497-t003:** Individual anthocyanins and antioxidant capacity of pomegranate juice samples.

Anthocyanin Composition (mg L^−1^)	AJ *	HPJ	MAJ
Dp3,5 **	182 ± 6.9 ^a^	175 ± 6.9 ^b^	93.9 ± 3.9 ^c^
Cya3,5	353 ± 1.9 ^a^	314 ± 3.5 ^b^	275 ± 1.3 ^c^
Dp3	9.8 ± 0.7 ^b^	13.3 ± 0.6 ^a^	4.9 ± 0.3 ^c^
Pg3,5	14.5 ± 0.7 ^a^	13.2 ± 0.4 ^b^	10.4 ± 0.3 ^c^
Cya3	24.6 ± 0.6 ^b^	30.8 ± 1.2 ^a^	16.7 ± 0.2 ^c^
Pg3	9.3 ± 0.5 ^a^	8.2 ± 0.4 ^b^	6.9 ± 0.1 ^c^
**Total**	**593 ± 1.9**	**555 ± 2.2**	**408 ± 1.0**
**Antioxidant capacity (mmol Trolox L^−1^)**			
DPPH^−^	10.1 ± 0.2 ^b^	14.7 ± 0.3 ^a^	5.8 ± 0.4 ^c^
ABTS^+^	11.3 ± 0.5 ^a^	16.6 ± 0.4 ^b^	6.1 ± 0.3 ^c^

* AJ—arils juice, HPJ—halved pomegranate juice, MAJ—macerated arils juice. Different letters (a, b, c) on the numbers in same row indicate significant differences (*p* < 0.05) between the juices. ** Dp3,5: delphinidin-3,5-diglucoside; Cya3,5: cyanidin-3,5-diglucoside; Dp3: delphinidin-3-glucoside; Pg3,5: pelargonidin-3,5-diglucoside; Cya3: cyanidin-3-glucoside, Pg3: pelargonidin-3-glucoside.

**Table 4 foods-10-01497-t004:** Volatile compounds of pomegranate juice samples (µg L^−1^).

Compounds	LRI *	AJ **	HPJ	MAJ	Identification ***
**Alcohols**					
Isobutanol	1094	5.5 ± 0.2 ^c^	16.7 ± 0.1 ^a^	13.0 ± 0.4 ^b^	LRI, MS, Ten
1-Butanol	1113	ND	ND	6.4 ± 0.1	LRI, MS, Std
1-Penten-3-ol	1157	5.9 ± 0.1 ^b^	4.4 ± 0.1 ^c^	9.8 ± 0.5 ^a^	LRI, MS, Std
Isoamyl alcohol	1221	10.2 ± 0.1 ^c^	38.0 ± 0.3 ^a^	32.8 ± 0.6 ^b^	LRI, MS, Std
1-Pentanol	1249	3.2 ± 0.2 ^a^	2.5 ± 0.1 ^a^	2.9 ± 0.1 ^a^	LRI, MS, Std
4-Heptanol	1272	4.5 ± 0.2 ^a^	1.4 ± 0.1 ^b^	1.9 ± 0.1 ^b^	LRI, MS, Std
2-Hexanol	1298	14.0 ± 0.3 ^a^	13.0 ± 0.2 ^b^	12.1 ± 0.1 ^c^	LRI, MS, Std
(*Z*)-2-Penten-1-ol	1320	6.6 ± 0.6 ^b^	3.6 ± 0.1 ^c^	9.6 ± 0.1 ^a^	LRI, MS, Std
1-Hexanol	1350	185 ± 3.2 ^b^	53.5 ± 1.1 ^c^	508 ± 6.4 ^a^	LRI, MS, Std
(*E*)-3-Hexen-1-ol	1371	9.1 ± 0.28 ^b^	7.3 ± 0.2 ^c^	15.0 ± 0.3 ^a^	LRI, MS, Std
(*Z*)-3-Hexen-1-ol	1378	147 ± 1.3 ^b^	123 ± 1.2 ^c^	352 ± 3.6 ^a^	LRI, MS, Std
3-Octanol	1393	ND	3.1 ± 0.1	ND	LRI, MS, Std
3-(Methylthio)-1-propanol	1710	ND	22.8 ± 0.7	12.0 ± 0.1	LRI, MS, Ten
2-Phenylethanol	1925	10.1 ± 0.1 ^c^	23.9 ± 0.4 ^b^	31.6 ± 0.5 ^a^	LRI, MS, Std
**Subtotal**		**401 ± 3.2**	**313 ± 2.2**	**1007 ± 6.8**	
**Ketones**					
Methylacetoin	1246	3.1 ± 0.1 ^a^	2.3 ± 0.1 ^b^	1.9 ± 0.1 ^c^	LRI, MS, Std
Acetoin	1292	24.9 ± 1.3 ^s^	93.9 ± 2.1 ^a^	42.3 ± 0.6 ^b^	LRI, MS, Std
2-Nonanone	1380	ND	1.43 ± 0.12	ND	LRI, MS, Std
2,6-Di(t-butyl)-4-hydroxy-4-methyl-2,5-cyclohexadien-1-one	2117	126 ± 1.2 ^c^	204 ± 3.2 ^b^	405 ± 2.3 ^a^	LRI, MS, Ten
**Subtotal**		**154 ± 2.1**	**302 ± 4.6**	**449 ± 5.1**	
**Acids**					
Pentanoic acid	1734	ND	14.2 ± 0.2	7.0 ± 0.1	LRI, MS, Std
Hexanoic acid	1834	13.8 ± 0.2	8.0 ± 0.1	ND	LRI, MS, Std
2-Ethylhexanoic acid	1968	ND	26.0 ± 1.1	60.0 ± 1.4	LRI, MS, Std
Octanoic Acid	2032	ND	4.1 ± 0.1	2.7 ± 0.01	LRI, MS, Std
**Subtotal**		**13.8 ± 0.2**	**52.3 ± 1.4**	**69.7 ± 1.8**	
**Phenols**					
3,4-Dimethylphenol	2189	ND	4.0 ± 0.1	5.3 ± 0.1	LRI, MS, Std
3-Ethylphenol	2195	ND	12.4 ± 0.2	10.9 ± 0.1	LRI, MS, Std
**Subtotal**		**ND**	**16.4 ± 0.2**	**16.2 ± 0.4**	
**Terpenes**					
Linalool	1530	ND	ND	2.1 ± 0.1	LRI, MS, Std
*α*-Terpineol	1688	18.8 ± 0.2 ^b^	15.8 ± 0.1 ^c^	23.7 ± 0.3 ^a^	LRI, MS, Std
*p*-Cymen-8-ol	1864	TR	1.4 ± 0.1	1.8 ± 0.1	LRI, MS, Std
**Subtotal**		**18.8 ± 0.2**	**17.2 ± 0.2**	**27.6 ± 0.1**	
**Benzene derivatives**					
Benzothiazole	1958	26.5 ± 0.1 ^c^	30.8 ± 0.4 ^b^	73.2 ± 2.2 ^a^	LRI, MS, Std
2-(Methylthio)benzothiazole	2422	ND	9.1 ± 0.1	4.7 ± 0.1	LRI, MS, Std
**Subtotal**		**26.5 ± 0.6**	**39.9 ± 0.3**	**77.9 ± 1.1**	
**Esters**					
(*Z*)-3-Hexenyl acetate	1305	ND	1.9 ± 0.1	ND	LRI, MS, Std
Isopulegol acetate	1608	6.6 ± 0.3 ^a^	2.2 ± 0.1 ^c^	3.7 ± 0.1 ^b^	LRI, MS, Std
**Subtotal**		**6.6 ± 0.3**	**4.1 ± 0.12**	**3.7 ± 0.1**	
**Aldehydes**					
n-Nonanal	1384	8.5 ± 0.2	ND	ND	LRI, MS, Std
**Subtotal**		**8.5 ± 0.2**	ND	ND	
**Norisoprenoids**					
*β*-Ionone	1947	TR	TR	28.9 ± 0.1	LRI, MS, Std
**Subtotal**		**TR**	**TR**	**28.9 ± 0.1**	
**Lactones**					
*γ*-Butyrolactone	1609	81.5 ± 1.1 ^b^	6.9 ± 0.1 ^c^	192 ± 0.2 ^a^	LRI, MS, Std
**Subtotal**		**81.5 ± 1.1**	**6.9 ± 0.1**	**192 ± 0.2**	
**Total**		**710.7 ± 7.9**	**751.8 ± 6.3**	**1872.0 ± 5.9**	

* LRI: Linear retention index calculated on DB-WAX capillary column; ** AJ—Arils juice, HPJ—halved pomegranate juice, MAJ—macerated arils juice); Results are the means of three repetitions as µg·L^−1^; *** Identification: Methods of identification; LRI (linear retention index), MS ten, (tentatively identified by MS), Std (chemical standard). When only MS or LRI is available for the identification of a compound, it must be considered as an attempt of identification. TR: Trace amount, ND: Not detected. Different letters (a, b, c) on the numbers in same row indicate significant differences (*p* < 0.05) between the juice samples. The bolds represent the subtotal and total amounts of each volatile group.

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
