# Peer review of "Elucidation of Volatiles, Anthocyanins, Antioxidant and Sensory Properties of cv. Caner Pomegranate (Punica granatum L.) Juices Produced from Three Juice Extraction Methods"

_foods, 2021, doi:10.3390/foods10071497_

Round 1

Reviewer 1 Report

REVIEW

It is a paper of moderate importance and low quality. First the English language needs correction by a native speaker or a person very fluent in Fnglish and I have many remarks which are listed here down

Line 39 function is th correct word and not operate

Line 41 ‘’several studies displayed’’ has to be replaced by “numerous studies demonstrated”

Lines 41-43 “that there is a  desired relation between inhibitions of many diseases especially cancers and the intake of  fruits and vegetables comprising natural antioxidants”  the English language has to be improved

Line 44 “enormous” must be replaced by “ substantial”

Line 51 use a better word than “dissimilar”

Line 53-55 “Even if exhaustive works are present in the literature concerning the health 53 effects of pomegranates and pomegranate juices, volatile compounds are less investi-54 gated.” English language has to be corrected

Lines 65-67 “Pomegranates are very rich in terms of anthocyanins, ellagic acid, 65 phytoestrogenic flavonoids and tannins which have the ability to act as antioxidant 66 properties … I think that you emphasize in antioxidant properties but you need to put some literature about the antimicrobial and other bioactive properties of pomegranate products

Line 80  extant is extent?

Line 90 “dried” you must clarify that you mean dried from the surface water to avoid confusion with real drying for all fruit

Line 132 What is the reason of determining both DPPH and ABTS? Please explain

Lines 141-151 Please give an appropriate reference for the extraction method because according to my knowledge much more advanced methods are used for this extraction

Line 152-165 you must clarify why you did not use standards for the quantification which is more accurate

Line 188-189 “The highest brix value was recorded at juices from the HPJ (16.0 °Brix) while the lowest was obtained from the MAJ (14.0 °Brix) sample.” Did you filter the juice before the Brix measurements because if not then maybe this is an error due to turbidity in addition the conclusion that tannins create this difference is not obvious because there is not analytical evidence for this. Please explain or remove this declaration

Line 209 please explain why “Hence, the use of juices with low values of L* parameter is rec ommended [28,29].” For example in tomato paste the high L value is desired. I really do not understand the claim that we do not want high L because anyway we will have oxidation

Table 3 how can you explain that the DPPH value of HPJ is higher than this of MAJ despite the fact that the HPJ is rich in tannins

Table 4. you must explain better the differences between the compositions of volatiles for the three treatments. Please find more literature for similar treatments and try to compare. For example if you see the terpen concentration, the highest figure corresponds to MAJ treatment which is more finished than the other two so did you use homogenization in order to have uniform particle size and compare better the quantified values of aroma constiuents.

How for example you could explain the big difference between the very similar treartments  AJ  and MAJ which do not have particles from the peel?

As you understand there are many issues to be solved and the most important issue is that the authors did not apply freezing before extraction in order to avoid the degradation of the valuable antioxidant and volatile constituents during the extraction of the juice.

For the above reason my recommendation is major revision and reconsideration.

Author Response

REVIEWER #1:

It is a paper of moderate importance and low quality. First the English language needs correction by a native speaker or a person very fluent in Fnglish and I have many remarks which are listed here down.

     Thank you for these remarks.

-Line 39 function is the correct word and not operate

The word “operate” was changed to “function”.

- Line 41 ‘’several studies displayed’’ has to be replaced by “numerous studies demonstrated”

Done as suggested; please, see line 40.

-Lines 41-43 “that there is a  desired relation between inhibitions of many diseases especially cancers and the intake of  fruits and vegetables comprising natural antioxidants”  the English language has to be improved.

The sentence was rephrased as “there has been a relation between the intake of fruits and vegetables comprising natural antioxidants and the inhibitions of many diseases including cancers.”, please see lines 41-42.

-Line 44 “enormous” must be replaced by “ substantial”

Done as suggested; please, see line 43.

-Line 51 use a better word than “dissimilar”

Done as suggested; please, see line 50.

-Line 53-55 “Even if exhaustive works are present in the literature concerning the health 53 effects of pomegranates and pomegranate juices, volatile compounds are less investi-54 gated.” English language has to be corrected.

The sentence was rephrased as “Even if exhaustive works are present in the literature concerning the health effects of pomegranates and their juices, only a small portion of them focus on volatile com-pounds.”, please see lines 52-54.

- Lines 65-67 “Pomegranates are very rich in terms of anthocyanins, ellagic acid, 65 phytoestrogenic flavonoids and tannins which have the ability to act as antioxidant 66 properties … I think that you emphasize in antioxidant properties but you need to put some literature about the antimicrobial and other bioactive properties of pomegranate products.

Authors thank the reviewer for this remark and the explanation and addition were done as suggested, please see lines 66-73.

-Line 80  extant is extent?

The word “extant” was changed to “extent”, please see line 81.

- Line 90 “dried” you must clarify that you mean dried from the surface water to avoid confusion with real drying for all fruit.

Reviewer is right, the explanation was added, please see line 91-92.

-Line 132 What is the reason of determining both DPPH and ABTS? Please explain.

The explanation of selection two methods was added to the results section of antioxidant analysis in lines 293-297.

-Lines 141-151 Please give an appropriate reference for the extraction method because according to my knowledge much more advanced methods are used for this extraction.

The liquid-liquid extraction procedure was applied to extract volatiles from pomegranate juices. The method was optimized according to authors’ previous study (reference 20, Selli and Kelebek, 2011). More details are available in this paper. Additionally, the explanation was added to lines 145-146.

- Line 152-165 you must clarify why you did not use standards for the quantification which is more accurate.

The identification of volatiles was supported by the standard use and used standards were displayed in Table 4 in “Identification” column. For the quantification, the internal standard method was preferred as the efficiency of this method was proven before in many other studies. Also, 4-nonanol was selected as an internal standard because of its high recovery.

-Line 188-189 “The highest brix value was recorded at juices from the HPJ (16.0 °Brix) while the lowest was obtained from the MAJ (14.0 °Brix) sample.” Did you filter the juice before the Brix measurements because if not then maybe this is an error due to turbidity in addition the conclusion that tannins create this difference is not obvious because there is not analytical evidence for this. Please explain or remove this declaration.

Authors thank the reviewer for this remark. The juices were filtered before the brix determination to avoid any errors. The difference in the values is thought to be caused for the rupture of the peel or the other internal tissues and passage of them into the juices. The explanation can be seen in lines 194-197.

-Line 209 please explain why “Hence, the use of juices with low values of L* parameter is recommended [28,29].” For example in tomato paste the high L value is desired. I really do not understand the claim that we do not want high L because anyway we will have oxidation.

In the literature, it was recommended for the pomegranate juices to have lower L* values that in case of oxidation, not much differences can be seen. The sentence was rephrased to clear the misunderstanding, please see lines 215-216.

-Table 3 how can you explain that the DPPH value of HPJ is higher than this of MAJ despite the fact that the HPJ is rich in tannins.

HPJ is richer in phenolic compounds, especially in anthocyanins (Table 3), than MAJ, the DPPH value of HPJ is expected to be higher than MAJ as there are more compounds to react with radical.

- Table 4. you must explain better the differences between the compositions of volatiles for the three treatments. Please find more literature for similar treatments and try to compare. For example if you see the terpen concentration, the highest figure corresponds to MAJ treatment which is more finished than the other two so did you use homogenization in order to have uniform particle size and compare better the quantified values of aroma constiuents.

Authors thank the reviewer for this remark. The maceration duration had an increasing effect on the terpene concentration in juices. Similar results were reported in macerated grapes before (Baron et al., 2017). Also, the homogenization of each juice sample was applied before the extraction to inhibit any impurities and get a better comparison between samples.

- How for example you could explain the big difference between the very similar treartments  AJ  and MAJ which do not have particles from the peel?

For the maceration process, pomegranate arils were crushed and macerated being in contact with the seeds for 3 hours (please see lines 95-100). For AJ, the seeds were immediately removed from the juice during squeezing. So both the exposure to the air and the contact with the seeds are being thought to be the reason of these differences we found.

- As you understand there are many issues to be solved and the most important issue is that the authors did not apply freezing before extraction in order to avoid the degradation of the valuable antioxidant and volatile constituents during the extraction of the juice.

Authors thank the reviewer for all remarks. All the suggestions were added in the manuscript accordingly.

Reviewer 2 Report

The manuscript entitled “Elucidation of volatiles, anthocyanins, antioxidant and sensory properties of cv. Caner (Punica granatum L.) pomegranate juices produced from three squeezing methods” reports a comprehensive work on the colour quality, volatiles, anthocyanins, sugar and organic acid constituents of cv. Caner pomegranate juice produced from different squeezing methods in the existing literature. This manuscript should have minor revisions.

The authors should use the same units, sometimes appears ml other mL.

All abbreviations should be reported for the first time.

In English the decimal is a point instead a comma, and the authors should pay attention to significant number.

Table 4. What is the meaning of TR and ND? Total should be subtotal, whereas general total should be Total.

Author Response

REVIEWER #2:

The manuscript entitled “Elucidation of volatiles, anthocyanins, antioxidant and sensory properties of cv. Caner (Punica granatum L.) pomegranate juices produced from three squeezing methods” reports a comprehensive work on the colour quality, volatiles, anthocyanins, sugar and organic acid constituents of cv. Caner pomegranate juice produced from different squeezing methods in the existing literature. This manuscript should have minor revisions.

-The authors should use the same units, sometimes appears ml other mL.

Authors thank the reviewer for this remark, all the units were check and homogenized throuout the text.

All abbreviations should be reported for the first time.

Authors thank the reviewer for this remark, all abbreviations were checked and explained.

In English the decimal is a point instead a comma, and the authors should pay attention to significant number.

All numeration was checked in both the tables and in the text.

Table 4. What is the meaning of TR and ND? Total should be subtotal, whereas general total should be Total.

The meanings of TR and ND were explained in line 404. Also, the totals were changed as suggested. 

Reviewer 3 Report

The manuscript reports the analytical characterization of bioactive molecules, and volatiles of three juice of pomegranate Caner cultivar, and determination of antioxidant activity and sensory properties. The article is of certain interest but the novelty should be more emphasized in both introduction and conclusion sections.

Introduction is well organized providing the reader a good state of art. Results and discussion are well presented. Also the experimental design has been well performed. In order to improve the quality of the present research article, could be of great interest give to the readers more information on the choice of the three juice extraction. Why the authors decide to adopt those three method of squeezing?

Moreover, are the analytical methods employed for this research paper previously used/validated? Could the authors specify this aspect? And in any case authors should add all information regarding linearity, recovery, accuracy and Limit of Detection and Limit of Quantification of the analytical methods employed.

There are some typos in the text, please check.

Author Response

REVIEWER #3:

The manuscript reports the analytical characterization of bioactive molecules, and volatiles of three juice of pomegranate Caner cultivar, and determination of antioxidant activity and sensory properties. The article is of certain interest but the novelty should be more emphasized in both introduction and conclusion sections.

Introduction is well organized providing the reader a good state of art. Results and discussion are well presented. Also the experimental design has been well performed. In order to improve the quality of the present research article, could be of great interest give to the readers more information on the choice of the three juice extraction. Why the authors decide to adopt those three method of squeezing?

Authors thank the reviewer for this remark. The explanation was added accordingly in lines 79-82.

Moreover, are the analytical methods employed for this research paper previously used/validated? Could the authors specify this aspect? And in any case authors should add all information regarding linearity, recovery, accuracy and Limit of Detection and Limit of Quantification of the analytical methods employed.

The applied methods were previously used by the authors and validated. The necessary information is added to lines 124-126 and 139-140.

There are some typos in the text, please check.

All the text was proofread again, please see the revisions in blue color.

Round 2

Reviewer 1 Report

Despite the fact most of my remarks have been corrected there is something important and is a big mistake. I give the paragraph were the reply is completely wrong: 

Table 3 how can you explain that the DPPH value of HPJ is higher than this of MAJ despite the fact that the HPJ is rich in tannins.

HPJ is richer in phenolic compounds, especially in anthocyanins (Table 3), than MAJ, the DPPH value of HPJ is expected to be higher than MAJ as there are more compounds to react with radical.

when a material is reach in antioxidants it is expected to have low and no high ic50.dpph. So the point rmains and the explanation is not sound but erroneous and it remains to be explained correctly

Author Response

  • When a material is reach in antioxidants it is expected to have low and no high ic50.dpph. So the point rmains and the explanation is not sound but erroneous and it remains to be explained correctly

            RESPOND: The authors thank reviewer for his/her attention. This remark is totally true, but there seems a little confusion about the antioxidant activity results. The authors did not specifically determine IC50 (%50 inhibition) values, instead the results indicate total antioxidant activity of pomegranate juices and result in the present study was given as mmol Trolox/L. As the reviewer noted, the IC50 value is inversely proportional to the free radical scavenging activity/antioxidant property of the samples.

In addition, mmol L-1 Trolox was changed as mmol Trolox L-1 in Final version text (highlighted red).